# Evaluation of Granger Causality Measures for Constructing Networks from Multivariate Time Series

**DOI:** 10.3390/e21111080

**Published:** 2019-11-04

**Authors:** Elsa Siggiridou, Christos Koutlis, Alkiviadis Tsimpiris, Dimitris Kugiumtzis

**Affiliations:** 1Department of Electrical and Computer Engineering, Aristotle University of Thessaloniki, University Campus, Thessaloniki 54124, Greece; esingiri@auth.gr (E.S.); ckoutlis@iti.gr (C.K.); 2Information Technologies Institute, Centre of Research and Technology Hellas, Thessaloniki 57001, Greece; 3Department of Computer, Informatics and Telecommunications Engineering, International Hellenic University, Serres 62124, Greece; atsimpiris@teicm.gr

**Keywords:** Granger causality, causality networks, dimension reduction measures, multivariate time series

## Abstract

Granger causality and variants of this concept allow the study of complex dynamical systems as networks constructed from multivariate time series. In this work, a large number of Granger causality measures used to form causality networks from multivariate time series are assessed. These measures are in the time domain, such as model-based and information measures, the frequency domain, and the phase domain. The study aims also to compare bivariate and multivariate measures, linear and nonlinear measures, as well as the use of dimension reduction in linear model-based measures and information measures. The latter is particular relevant in the study of high-dimensional time series. For the performance of the multivariate causality measures, low and high dimensional coupled dynamical systems are considered in discrete and continuous time, as well as deterministic and stochastic. The measures are evaluated and ranked according to their ability to provide causality networks that match the original coupling structure. The simulation study concludes that the Granger causality measures using dimension reduction are superior and should be preferred particularly in studies involving many observed variables, such as multi-channel electroencephalograms and financial markets.

## 1. Introduction

Real-world complex systems have been studied as networks formed from multivariate time series, i.e., observations of a number of system variables, such as financial markets and brain dynamics [1,2]. The nodes in the network are the observed variables and the connections are defined by an interdependence measure. The correct estimation of the interdependence between the observed system variables is critical for the formation of the network and consequently the identification of the underlying coupling structure of the observed system.

Many interdependence measures that quantify the causal effect between the variables observed simultaneously in a time series are based on the concept of Granger causality [3]: a variable *X* Granger causes a variable *Y* if the information in the past of *X* improves the prediction of *Y*. The concept was first mentioned by Wiener [4], and it is also referred to as Wiener–Granger causality, but, for brevity, we use the common term Granger causality here. The methodology on Granger causality was first developed in econometrics, and it has been widely applied to many other fields, such as cardiology and neuroscience (analysis of electroencephalograms (EEG) and functional magnetic resonance imaging (fMRI) [5,6,7,8]) and climate [9,10].

The initial form of Granger causality based on autoregressive models has been extended to nonlinear models, basically local linear models [11,12,13,14], but also kernel-based and radial basis models [15,16,17,18], and recently more advanced models, such as neural networks [19,20]. In a wider sense, the directed dependence inherent in Granger causality is referred to as coupling, synchronization, connectivity and information flow depending on the estimation approach for the interdependence. Geweke [21,22] defined an analogue of Granger causality in the frequency domain, developed later to other frequency measures [23,24]. A number of nonlinear measures of interdependence inspired by the Granger causality idea have been developed, making use of state-space techniques [25,26], information measures [27,28,29], and techniques based on the concept of synchronization [30,31,32]. We refer to all these measures as causality measures (in the setting of multivariate time series) and the networks derived by these measures as causality networks.

Distinguishing indirect and direct causality with the available methods is a difficult task [33,34], and multivariate measures are expected to address better this task than bivariate measures. The bivariate causality measures do not make use of the information of other observed variables besides the variables *X* and *Y* of the examined causality from *X* to *Y*, denoted X→Y, and thus estimate direct and indirect causality, where indirect causality is mediated by a third variable *Z*, e.g., X→Z and Z→Y results in X→Y. The multivariate causality measures apply conditioning on the other observed variables to estimate direct causal effects, denoted as X→Y|Z, where *Z* stands for the other observed variables included as conditioning terms [35,36,37].

For high-dimensional time series, i.e., a large number *K* of observed variables, the estimation of direct causal effects is difficult and the use of multivariate causality measures is problematic. One solution to this problem is to account only for a subset of the other observed variables based on some criterion of relevance to the driving or response variable [38]. In a different approach, dimension reduction techniques have been embodied in the computation of the measure restricting the conditioning terms, and they have been shown to improve the efficiency of the direct causality measures [29,39,40,41,42,43].

Recent comparative studies have assessed causal effects with various causality measures, using also significance tests for each causal effect [14,44,45,46,47,48,49,50,51]. Some studies concentrated on the comparison of direct and indirect causality measures [46,52], whereas other studies focused on specific types of causality measures, e.g., frequency domain measures [53,54,55,56], or different significance tests for a causality measure [57,58,59]. These studies are done on specific real data types, mostly from brain, which limits the generalization of the conclusions.

Whereas most of the abovementioned studies were concentrated on the estimation of specific causal effects by the tested measures, this study is merely focused on assessing the bivariate (X→Y) and multivariate and (X→Y|Z) causality measures that estimate best the whole set of causal effects for all pairs (*X*,*Y*), i.e., the true causality network. In particular, high dimensional systems and subsequently high-dimensional time series are considered, so that the estimated networks have up to 25 nodes. Some first results of the application of different causality measures on simulated systems and evaluation of their accuracy in matching the original network were presented earlier in [60]. As the focus is on the preservation of the original causality network, we assess the existence of each causality term applying appropriate significance criteria. The causality measures are ranked related to their ability to match the original causality networks of different dynamical systems and stochastic processes. For the computation of the causality measures, several software programs are freely accessible [61,62,63,64,65,66], but we have developed most of the causality measures in the context of previous studies of our team, and few measures were run from the software [63].

The structure of the paper is as follows. In Section 2, we present the causality measures, the identification of network connections from each measure, the statistical evaluation of the accuracy of each measure in identifying the original coupling network, the formation of a score index for each measure, and finally the synthetic systems used in the simulations. In Section 3, we present the results of the measures on these systems, and we rank the measures as for their accuracy in matching the original coupling network. Discussion follows in Section 4, with conclusions presented in the final section.

## 2. Methodology

The methodology implemented for the comparative study of causality measures aiming at evaluating the measures and ranking them as to their accuracy in identifying correctly the underlying coupling network is presented here. The methodology includes the causality measures compared in the study, the techniques for the identification of network connections from each measure, the statistical evaluation of the accuracy of each measure in identifying the original coupling network, and the formation of an appropriate score index for the overall performance of each measure. Finally, the synthetic systems used in the simulation study are presented.

### 2.1. Causality Measures

First, it is noted that in this comparative study correlation or in general symmetric measures of *X* and *Y* are not considered. Many such measures were initially included in the study, e.g., many phase-based measures such as phase locking value (PLV) [67], phase lag index (PLI) [68] and weighted phase lag index (wPLI) [69], rho index (RHO) [70], phase slope index (PSI) [31], and mean phase coherence (MPC) [71]. However, their evaluation in the designed framework is not possible as the identification of the exact directed connections of the original coupling network is quantified to assess the measure performance.

Causality measures can be divided in three categories as to the domain of data representation they are defined in: time, frequency, and phase (see Figure 1).

The measures in time domain dominate and they are further divided in model-based and model-free measures. Many of the model-free measures are based on information theory measures and the other model-free measures on the time domain are referred to as “other” measures. Thus, five main classes of causality measures are considered in this study: model-based measures, information measures, frequency measures, phase measures, and other measures that cannot be defined in terms of the other four classes. The measures organized in these five classes and included in the comparative study are briefly discussed below, and they are listed in Table 1, with reference and code number denoting the type of measure (the class, bivariate or multivariate and with our without dimension reduction).

The first class of model-based measures regards measures that implement the original concept of Granger causality, the bivariate measure of Granger causality index (GCI) [3] (only *X* and *Y* variables are considered), and the multivariate measures of the conditional Granger causality index (CGCI) [22] and the partial Granger causality (PGCI) [72] (also the other observed variables denoted *Z* are included). All these measures require the fit of a vector autoregressive model (VAR), on the two or more variables. The order *p* of VAR denoting the lagged variables of each variable contained in the model can be estimated by an information criterion such as the Akaike information criterion (AIC) and the Bayesian Information Criterion (BIC), which often does not provide optimal lags, e.g., see the simulation study in [83] and citations therein, and the so-called *p*-hacking (in the sense of *p*-value) in terms of VAR models for Granger causality in [84]. To overcome the use of order estimation criteria, here we use a couple of predefined appropriate orders *p* for each system (see Table 1). In the presence of many observed variables, dimension reduction in VAR has been proposed, and here we use the one developed from our team, the restricted conditional Granger causality index (RCGCI) [43]. Thus, for this class of measures, there are bivariate and multivariate measures, and multivariate measures that apply dimension reduction, as noted in the sketched division of causality measures in Figure 1. These are all linear measures and besides this constraint they have been widely used in applications. Other nonlinear extensions are not considered in this study for two reasons: either they were very computationally intensive, such as the cross predictions of local state space models [14], or they were too complicated so that discrepancies to the original methods may occur [19,20].

The information measures of the second class have also been popular in diverse applications recently due to their general form, as they do not require any specific model, they are inherently nonlinear measures and can be applied to both deterministic systems and stochastic processes of any type, e.g., oscillating flows and discrete maps of any dimension. The main measure they rely on is the mutual information (MI), and more precisely the conditional mutual information (CMI). There have been several forms for causality measures based on MI and CMI in the literature, e.g., see the coarse-grained mutual information in [85], but the prevailing one is the transfer entropy (TE), originally defined for two variables [27]. The multivariate version, termed partial transfer entropy (PTE) was later proposed together with different estimates of the entropies involved in the definition of PTE, bins [86], correlation sums [87], and nearest neighbors [73]. Here, we consider the nearest neighbor estimate for both TE and PTE, found to be the most appropriate for high dimensions. Equivalent forms to TE and PTE are defined for the ranks of the embedding vectors rather than the observations directly. We consider the bivariate measures of symbolic transfer entropy (STE) [74] and transfer entropy on rank vectors (TERV) [76], and the multivariate measures of partial symbolic transfer entropy (PSTE) [75] and partial transfer entropy on rank vectors (PTERV) [77]. The idea of dimension reduction was implemented in TE first, applying a scheme for a sparse non-uniform embedding of both *X* and *Y*, termed mutual information on mixed embedding (MIME) [29]. This bivariate measure was later extended to the multivariate measure of partial MIME (PMIME) [42]. Only the PMIME is included in the study simply due to the computational cost, and it is noted that by construction the measure gives zero for insignificant causal effects, so it does not require binarization when networks of binary connections have to be derived (the positive values are simply set to one).

All the methods in the third class of frequency measures rely on the estimation of the VAR model, either on only the two variables *X* and *Y* or on all the observed variables (we consider only the latter case here). Geweke’s spectral Granger causality (GGC) is the early measure implementing the concept of Grangre causality in the frequency domain [21,80], included in the study. Another older such measure included in the study is the direct transfer function (DTF) [23], which, although it is a multivariate measure, it does not discriminate direct from indirect causal effects. For this, an improvement is proposed and used also in this study, termed direct directed transfer function (dDTF) [79]. We also consider the partial directed coherence (PDC) [24] and the improved version of generalized partial directed coherence (GPDC) [78], which have been particularly popular in EEG analysis. When applied to EEG, the measures are defined in terms of frequency bands of physiological relevance (δ, θ, α, β, γ, going from low to high frequencies), and the same proportional split of the frequency range is followed here (as if the sampling frequency was 100 Hz). Finally, we consider a dimension reduction of VAR in the GPDC measure called restricted GPDC (RGPDC), recently proposed from our team [81].

As mentioned above the class of phase measures contains a good number of measures used in connectivity analysis, mainly in neuroscience dealing with oscillating signals such as EEG, but most of these measures are symmetric and thus out of the scope of the current study. In the evaluation of the causality measures, we consider the bivariate measure of phase directionality index (DPI) [30], which is a measure of synchronization designed for oscillating time series. Information measures have also been implemented in the phases, e.g., see [88], but not considered here.

Another class of measures used mainly in neuroscience regards inter-dependence measures based on neighborhoods in the reconstructed state space of each of the two variables *X* and *Y*. A series of such measures have been proposed after the first work in [25], using also ranks of the reconstructed vectors [89], the latter making the measure computationally very slow. The convergent cross mapping is developed under the same approach [90], and the same yields for the measure of mean conditional recurrence (MCR) [26]. It is noted that all these measures are bivariate and they are expected to suffer from estimating indirect connections in the estimated causality network. The MCR is included as a representative of the state space bivariate measures in the class of other measures. In this class, we include also event synchronization measures [82], and specifically the direct event delay (DED) that is a directional bivariate measure, considered as causality measure and included in the study.

For the information measures where the estimation of entropies in high dimensions is hard, the comparison of the multivariate measures that do not include dimension reduction to these including dimension reduction would be unfair when high-dimensional systems are considered. To address this, in the calculation of a multivariate information measure not making use of dimension reduction, we choose to restrict the set of the conditioning variables *Z* in the estimated causal effect X→Y|Z to only the three more relevant variables. In the simulations, we consider the number of observed variables (equal to the subsystems being coupled) to be K=5 and K=25, while, for K=5, all the remaining variables are considered in *Z*; for K=25, only three of the remaining 23 variables are selected. The criterion of selection is the mutual information of the remaining variables to the driving variable *X*, which is common for the selection of variables [38].

### 2.2. Identification of Original Network Connections

We suppose that a dynamical system is formed by the coupling of *K* subsystems, and we observe one variable from each subsystem, so that a multivariate time series of dimension *K* is derived. The coupling structure of the original system can be displayed as a network of *K* nodes where the connections are determined by the system equations. Formally, in the graph-theoretical framework, a network is represented by a graph G=(V;E), where *V* is the set of *K* nodes, and *E* is the set of the connections among the nodes of *V*. The original coupling network is given as a graph of directed binary connections, where the connection from node *i* to node *j* is assigned with a value $a_{i,j}$ being one or zero, depending whether variables of the subsystem *i* are present in the equation determining the variables of the subsystem *j*. The components $a_{i,j}$, i,j=1,…,K, form the adjacency matrix *A* that defines the network.

The computation of any causality measure presented in Section 2.1 on all the directed pairs (i,j) of the *K* observed variables gives a weight matrix *R* (assuming only positive values of the measure, so that a transformation of the measure can be applied if necessary). Thus, the pairwise causality matrix *R* with entries $R_{i,j}\;=\;R_{X_{i}\to X_{j}}$ defines a network of weighted connections, assigning the weighted directed connection $R_{i,j}$ from each node *i* to each node *j*.

In applications, often binary networks are sought to better represent the estimated structure of the underlying system. Here, we are interested to compare how the causality measure retrieves the original directed coupling structure, and therefore we want to transform the weighted network to a binary network. Commonly, the weighted matrix *R* is transformed into an adjacency matrix *A* by suitable thresholding, keeping in the graph only connections with weights higher than some threshold (and setting their weights to one) and removing the weaker connections (setting their weights to zero). For each causality measure, an appropriate threshold criterion is sought to determine the significant values of the measure that correspond to connections in the binary network. We consider three approaches for thresholding that have been used in the literature:Rigorous thresholding is provided by an appropriate significance test for the causality measure $R_{X_{i}\to X_{j}}$. For all considered causality measures in this study, we expect the causality measure to lie at the zero level if there is no causal effect and be positive if it is, so that the test is one-sided. Thus, the null and alternative hypotheses are respectively:
(1)$$H_{0}:R_{X_{i}\to X_{j}}=0,\;\;H_{1}:R_{X_{i}\to X_{j}}>0.$$Parametric significance tests have been developed only for the linear causality measures, and for consistency we apply the randomization significance test to all causality measures, making use of the simple technique of time-shifted surrogates. Specifically, we generate *M* resampled (surrogates) time series for the driving variable *X*, each by shifting cyclically the original observations of *X* by a random step *w*. For the original time series of the driving variable $X_{i}$ denoted $\left\{X_{i,t}\right\}\;=\;\left\{x_{1,t},x_{2,t},\ldots ,x_{n,t}\right\}$, the surrogate time series is $\left\{X_{i,t}^{*}\right\}\;=\;\left\{x_{w+1,t},x_{w+2,t},\ldots ,x_{n,t},x_{1,t},\ldots ,x_{w-1,t},x_{w,t}\right\}$. In this way, we destroy any form of coupling of $X_{i}$ and any other variable $X_{j}$, so that $\left\{X_{i,t}^{*}\right\}$ is consistent to $H_{0}$, but it preserves the dynamics and the marginal distribution of $X_{i}$. The test statistic is the causality measure $R_{X_{i}\to X_{j}}$, and it takes the value $R_{0}$ on the original time series pair and the values $R_{1},R_{2},\ldots ,R_{M}$ on each of the *M* resampled time series pairs. The rank of $R_{0}$ in the ordered list of M+1 values $R_{0},R_{1},R_{2},\ldots ,R_{M}$, denoted $r_{0}$, defines the *p*-value of the randomization test as $p\;=\;1-\frac{r_{0}-0.326}{M+1+0.348}$ [91]. If $R_{0}$ is at the right tail of the empirical distribution formed by $R_{1},R_{2},\ldots ,R_{M}$, then the H${}_{0}$ is rejected, which suggests that p<α, where α is the significance level of the test determining the tail. For a multivariate time series of *K* variables, K(K−1) significance tests in total are performed for each causality measure, indicating that multiple tests are performed on the same dataset. This is a known issue in statistics and corrections for multiple testing can be further be applied, such as the false discovery rate [92]. Here, we refrain from using such a correction and rather use three different significance levels α=0.01,0.05,0.1. We opted for this as the same setting is applied for all causality measures.The second thresholding criterion is given by the desired density of the binary network. In the simulation study, we know the density of the original network, denoted by the number of connections $\rho_{0}$. We consider five different values for the density ρ of the estimated causality binary network given in multiples of $\rho_{0}$ as 0.6,0.8,1.0,1.2,1.4.The third thresholding criterion is simply given by a predefined magnitude threshold on the causality measure. Here, we select an appropriate threshold $th_{\rho }$ separately for each causality measure and each coupling strength for the same system, where ρ indicates the corresponding density. Having 10 realizations for each scenario (system and coupling strength), the $th_{\rho }$ is the average of the thresholds found for the given density ρ in the 10 realizations.

### 2.3. Performance Indices for Statistical Evaluation of Methods Accuracy

For a system of *K* variables, there are K(K−1) ordered pairs of variables to estimate causality. In the simulations of known systems, we know the true coupling pairs and thus we can compute performance indices to rate the causality measures as for their overall matching of the original connections in the network. Here, we consider the performance indices of specificity, sensitivity, Matthews correlation coefficient, F-measure, and Hamming distance.

The sensitivity is the proportion of the true causal effects (true positives, TP) correctly identified as such, given as sens = TP/(TP + FN), where FN (false negatives) denotes the number of pairs having true causal effects but have gone undetected. The specificity is the proportion of the pairs correctly not identified as having causal effects (true negatives, TN), given as spec = TN/(TN + FP), where FP (false positives) denotes the number of pairs found falsely to have causal effects. An ideal causality measure would give values of sensitivity and specificity at one. To weigh sensitivity and specificity collectively, we consider the Matthews correlation coefficient (MCC) [93] given as
(2)$$\mathrm{MCC}=\frac{\mathrm{TP}\cdot \mathrm{TN}-\mathrm{FP}\cdot \mathrm{FN}}{\sqrt{\left(\mathrm{TP}+\mathrm{FP})\cdot (\mathrm{TP}+\mathrm{FN})\cdot (\mathrm{TN}+\mathrm{FP})\cdot (\mathrm{TN}+\mathrm{FN}\right)}}.$$
MCC ranges from −1 to 1. If MCC = 1, there is perfect identification of the pairs of true and no causality, if MCC = −1, there is total disagreement and pairs of no causality are identified as pairs of causality and vice versa, whereas MCC at the zero level indicates random assignment of pairs to causal and non-causal effects.

Similarly, we consider the F-measure that combines precision and sensitivity. The precision, called also positive predictive value, is the number of detected true causal effects divided by the total number of detected casual effects, given as prec = TP/(TP + FP), and the F-measure (FM) is defined as
$$\mathrm{FM}=\frac{2\cdot \mathrm{prec}\cdot \mathrm{sens}}{\mathrm{prec}+\mathrm{sens}}=\frac{2\mathrm{TP}}{2\mathrm{TP}+\mathrm{FN}+\mathrm{FP}},$$
which ranges from 0 to 1. If FM = 1, there is perfect identification of the pairs of true causality, whereas, if FM = 0, no true coupling is detected.

The Hamming distance (HD) is the sum of false positives (FP) and false negatives (FN), HD = FP + FN. Thus, HD gets non-negative integer values bounded below by zero (perfect identification) and above by K(K−1) if all pairs are misclassified.

### 2.4. Score Index

In Section 2.3, we presented five performance indices to evaluate in different ways the match of the original network and the estimated network from each causality measure. Furthermore, we want to quantify this match for different settings, which involve different systems and different scenarios for each system regarding the number of variables *K*, the system free parameter, the time series length *n* and the coupling strength *C*, where applicable (to be presented below in Section 2.5). For this, we rely on the index MCC, and, for each scenario of a different system, we rank the causality measures according to their average MCC (across 10 realizations generated per scenario). For equal MCC, ordinal ranking (called also “1234” ranking) is adopted [94]. Specifically, the order of measures of equal MCC is decided from distinct ordinal numbers given at random to each measure of equal MCC value. Next, we derive the average rank of a causality measure *i* for all different coupling strengths *C* of a system *j*, $P_{i,j}$, as the average of the ranks of the causality measure *i* in all coupling strengths tested for the system *j*. A score $s_{i,j}$ of the causality measure *i* for the system *j* is then derived by normalization of the average rank $P_{i,j}$ over the number *N* of all measures so as to scale between zero and one, $s_{i,j}\;=\;\left(N-P_{i,j}\right)/\left(N-1\right)$, where one is for the best measure ranked at top. The overall score of the causality measure *i* over all systems, $s_{i}$, is simply given by the average $s_{i,j}$ over all systems, including the two different *K* values for systems S1, S2 and S3, the two values for the system parameter Δ and *A* for systems S2 and S3, respectively, and the two time series lengths *n* for system S1. The systems are presented in the next section.

### 2.5. Synthetic Systems

For the comparative study, we use four systems with different properties: the coupled Hénon maps as an example of discrete-time chaotic coupled system [42], the coupled Mackey–Glass system as an example of continuous-time chaotic coupled system but of high complexity [42,95], the so-called neural mass model as an example of a continuous-time stochastic system used as an EEG model [96,97], and the vector autoregressive model (VAR) as suggested in [98], used as an example of a discrete-time linear stochastic process. The four systems are briefly presented below:

**S1**: The system of coupled Hénon maps is a system of coupled chaotic maps defined as
(3)$$\;\begin{array}{cc} x_{i,t}=1.4-x_{i,t-1}^{2}+0.3x_{i,t-2}, & \mathrm{for}\;\;i=1,K,\\ x_{i,t}=1.4-{\left(0.5C\left(x_{i-1,t-1}+x_{i+1,t-1}\right)+\left(1-C\right)x_{i,t-1}\right)}^{2}+0.3x_{i,t-2}, & \mathrm{for}\;\;j=2,\ldots ,K-1, \end{array}$$
where *K* denotes the number of variables and *C* is the coupling strength. We consider the system for K=5 and K=25 and the corresponding coupling network is shown in Figure 2a,b, respectively.

Multivariate time series of size *K* are generated, and we use short and long time series of length n=512 and n=2048, respectively. An exemplary time series for K=5 is given in Figure 3a.

**S2**: The system of coupled Mackey–Glass is a system of coupled identical delayed differential equations defined as
(4)$${\dot{x}}_{j}\left(t\right)=-0.1x_{j}\left(t\right)+\sum_{i=1}^{K}\frac{C_{ij}x_{i}\left(t-\Delta \right)}{1+x_{i}{\left(t-\Delta \right)}^{10}}\;\;\mathrm{for}\;\;j=1,2,\ldots ,K,$$
where *K* is the number of subsystems coupled to each other, $C_{ij}$ is the coupling strength, and Δ is the lag parameter. We set $C_{ii}\;=\;0.2$ and $C_{i,j}$ for i≠j zero or *C* according to the ring coupling structure shown in Figure 2a,b for K=5 and K=25, respectively. For details on the solution of the delay differential equations and the generation of the time series, see [42]. Two scenarios are considered regarding the inherent complexity of each of the *K* subsystems given by Δ=100 and Δ=300, regarding high complexity (correlation dimension is about 7.0 [99]) and even higher complexity (not aware of any specific study for this regime), respectively. Exemplary time series for each Δ and K=5 are given in Figure 3b,c. The time series used in the study have length n=4096.

**S3**: The neural mass model is a system of coupled differential equations with a stochastic term that produces time series similar to EEG simulating different states of brain activity, e.g., normal and epileptic. It is defined as
(5)$$\begin{array}{cc} {\dot{y}}_{0}^{j}\left(t\right)= & y_{3}^{j}\left(t\right),\\ {\dot{y}}_{3}^{j}\left(t\right)= & AaS\left[y_{1}^{j}\left(t\right)-y_{2}^{j}\left(t\right)\right]-2ay_{3}^{j}\left(t\right)-a^{2}y_{0}^{j}\left(t\right),\\ {\dot{y}}_{1}^{j}\left(t\right)= & y_{4}^{j}\left(t\right),\\ {\dot{y}}_{4}^{j}\left(t\right)= & Aa\left\{p^{j}\left(t\right)+C_{2}S\left(C_{1}y_{0}^{j}\right)+\sum_{\begin{array}{c} i=1\\ i\neq j \end{array}}^{K}C_{ij}y_{6}^{i}\left(t\right)\right\},\\ \; & -2ay_{4}^{j}\left(t\right)-a^{2}y_{1}^{j}\left(t\right),\\ {\dot{y}}_{2}^{j}\left(t\right)= & y_{5}^{j}\left(t\right),\\ {\dot{y}}_{5}^{j}\left(t\right)= & Bb\left\{C_{4}S\left[C_{3}y_{0}^{j}\left(t\right)\right]\right\}-2by_{5}^{j}\left(t\right)-b^{2}y_{2}^{j}\left(t\right),\\ {\dot{y}}_{6}^{j}\left(t\right)= & y_{7}^{j}\left(t\right),\\ {\dot{y}}_{7}^{j}\left(t\right),= & Aa_{d}S\left(y_{1}^{j}\left(t\right)-y_{2}^{j}\left(t\right)\right)-2a_{d}y_{7}^{j}\left(t\right)-a_{d}^{2}y_{6}^{j}\left(t\right), \end{array}$$
where *j* denotes each of the *K* subsystems representing the neuron population defined by eight interacting variables and the population (subsystem) interacts with other populations through variable $y_{4}^{j}$ with coupling strength $C_{ij}$. We set $C_{ii}\;=\;0.0$ and $C_{i,j}$ for i≠j zero or *C* according to the ring coupling structure shown in Figure 2a,b for K=5 and K=25, respectively. The term $p^{j}\left(t\right)$ represents a random influence from neighboring or distant populations, *A* is an excitation parameter and *B*, *a*, *b*, $a_{d}$, C1–C4 other parameters (see [96] for more details). The function *S* is the sigmoid function $S\left(v\right)\;=\;2e_{0}/\left(1+e^{r(v_{0}-v)}\right)$, where *r* is the steepness of the sigmoid and $e_{0}$, $v_{0}$ are other parameters explained in [96]. From each population j=1,…,K, we consider only the first variable $y_{0}^{j}$ and obtain the multivariate time series of *K* variables. The value of the excitation parameter *A* affects the form of the output signals combined with the coupling strength level, ranging from similar to normal brain activity with no spikes to almost periodic with many spikes similar to epileptic brain activity. We consider two values for this parameter, one for low excitation with A=3.45 and one for high excitation with A=3.7. Exemplary time series for each *A* and K=5 are given in Figure 3d,e. The time series used in the study have length n=4096.

**S4**: The VAR process on K=25 variables and order P=3 as suggested in [98] is used as a representative of a high-dimensional linear stochastic process. Initially, 4% of the coefficients (total coefficients 1875) of VAR(3) selected randomly are set to 0.9 and the rest are zero and the positive coefficients are reduced iteratively until the stationarity condition is fulfilled. The autoregressive terms of lag one are set to one. The true couplings are 8% of a total of 600 possible ordered couplings. An exemplary coupling network of random type is shown in Figure 2c. The time series length is set to n=512 and an exemplary time series of only five of the K=25 variables of the VAR(3) process is shown in Figure 3f.

For S1, S2, and S3, the coupling strength *C*, fixed for all couplings, is varied to study a wide range of coupling levels from zero coupling to very strong coupling. Specifically, for S1 and S2, C=0,0.05,0.1,0.2,0.3,0.4,0.5, and for S3 C=0,20,40,80,120,160,200. For S4, only one case of coupling strength is considered, given by the magnitude 0.23 of all non-zero coefficients, for which the stationarity of VAR(3) process is reached. For each system and scenario of different coupling strength, 10 multivariate time series (realizations) are generated to obtain statistically valid results. The evaluation is performed as described in Section 2.4.

## 3. Results

In this section, the evaluation of the performance of all causality measures is presented for each system and scenario. First, the procedure of the evaluation is shown in one specific setting, then the measures are evaluated and ranked for each system and finally the overall ranking is given.

### 3.1. Evaluation of Measures in One Exemplary Setting

We consider a multivariate time series of length n=512 from the system S1 of coupled Hénon maps for K=5 variables and coupling strength C=0.2. The original coupling network has the ring structure as shown in Figure 2a. We derive the estimated causality (weight) matrix by the bivariate measure of transfer entropy (TE) using the appropriate parameters of embedding dimension m=2 and τ=1
(6)$$R_{\mathrm{TE}}=\left(\begin{array}{ccccc} 0 & 0.148 & -0.003 & -0.002 & -0.012\\ -0.005 & 0 & 0.167 & 0.013 & -0.021\\ -0.015 & 0.094 & 0 & 0.051 & -0.004\\ -0.015 & 0.016 & 0.092 & 0 & -0.014\\ -0.011 & -0.009 & 0.003 & 0.193 & 0 \end{array}\right),$$
where the negative values denote the negative bias in the estimation of TE with the nearest neighbors estimate. Applying the three criteria of measure significance for transforming the weight matrix to an adjacency matrix (see Section 2.2), we derive the causality binary networks. Specifically, as shown in Figure 4, different binary networks are obtained for the different values of the significance level of the randomization test, the network density, and the magnitude threshold.

In Figure 4a, the binary network for significance level α=0.01 and α=0.05 coincides with the original coupling network, whereas, for α=0.1, more connections are present regarding indirect causal effects (for the latter, the statistical significant values are given in bold in the form of Equation (Equation 6)). When preserving the four, six, and eight strongest connections, as shown in Figure 4b, the true structure is preserved only when the correct density is set (six connections), indicating that the highest TE values are reached at the true couplings. This seems the optimal strategy for thresholding, but, in real-world applications, the actual network density is not a priori known. Similarly, in Figure 4c, the binary networks obtained using three magnitude thresholds on TE values are shown. Each of the three magnitude thresholds is computed as the average threshold for preserving the corresponding network density across 10 realizations, for this specific coupling strength and causality measure. These magnitude thresholds happen not to be the ones corresponding to the network densities for this realization and actually none of the three thresholds identifies all the existing and non-existing connections.

For illustration purposes, we compute the performance indices for TE at this scenario using the statistical significance criterion for α=0.1, given in Table 2.

We note that the two extra connections found significant using α=0.1 reduce the specificity to 0.86 while sensitivity is one, which affects accordingly the other three measures. Note that the mismatch of just two out of 20 connections (HD=2) gives MCC=0.8, significantly lower from one, and the same holds for the F-measure index.

For the same scenario, the measure of PMIME (for a maximum lag L=5 well above the optimal lag 2) gives a weight matrix of zero and positive numbers
$$R_{\mathrm{PMIME}}=\left(\begin{array}{ccccc} 0 & 0.162 & 0 & 0 & 0\\ 0 & 0 & 0.101 & 0 & 0\\ 0 & 0.076 & 0 & 0.077 & 0\\ 0 & 0 & 0.094 & 0 & 0\\ 0 & 0 & 0 & 0.173 & 0 \end{array}\right).$$
No significance criterion is applied here, and simply setting the positive numbers to one gives the adjacency matrix, and, in this case, the estimated causality network matches exactly the original coupling network giving HD = 0 and all other indices equal to one.

### 3.2. Results with Respect to Performance Indices, Significance Criteria, and Coupling Strength

First, we give a comprehensive presentation reporting all the performance indices presented in Section 2.3 and the significance criteria in Section 2.2 for system S1 and K=25, n=2048 and C=0.2. In Table 3, the five performance indices of the eight highest ranked causality measures in terms of MCC are presented.

In Table 3 and all tables to follow, the measures making use of dimension reduction, i.e., PMIME, RCGCI, and RGPDC, are highlighted (bold face) to accommodate comparison with the other measures. The results are organized in three blocks, one for each of the three significance criteria. For the criterion of statistical significance at α=0.05, the dimension reduction measures score highest in all performance indices. The PMIME (L=5) measure obtains the greatest specificity value 0.99 and RCGCI (p=5) the greatest sensitivity value 0.92. A large difference between the first MCC = 0.86 for PMIME and the MCC for the other highest ranked measures is observed, while, for the specificity and sensitivity indices, this does not hold. This is explained by the fact that a small decrease in specificity implies increase in the number of falsely detected causal effects that, for networks of low density, dominates in the determination of MCC (see Equation (Equation 2)). For the significance criteria of density ($\rho_{0}\;=\;48$ equal to the number of true couplings) and threshold ($th_{48}$), PMIME (L=5) is unaltered at the first rank while the information and frequency measures exhibit better performance compared to the criterion of statistical significance. It is also observed that these two criteria show lower sensitivity and higher specificity, which questions the rule that the couplings of largest causality values are the true ones. When ρ is smaller or larger than the true density, sensitivity changes more than specificity again due to the sparseness of the true network. Similar conclusions are inferred for the significance criterion of magnitude threshold.

We demonstrate further the dependence of the causality measure performance on the parameter in each significance criterion using the S4 system of the VAR process (K=25, n=512, P=3). In Table 4, the ranking of the eight best measures in terms of MCC is presented for three parameters of each of the three significance criteria.

For this linear system, the highest ranked causality measures for all three significance criteria are the linear measures using dimension reduction RCGCI and RGPDC for various parameter values. This is somehow expected as these measures are both linear and the underlying system is linear, and they use dimension reduction as the number of variables is K=25. For such a high-dimensional time series, the bivariate linear measures give indirect (and false in our evaluation) causality effects, whereas the multivariate linear measures without dimension reduction cannot reach the performance of RCGCI and RGPDC as the time series length n=512 is relatively small for estimating accurately the VAR model parameters (75 coefficients in VAR(3) are to be estimated for each of the K=25 variables). The best performance of the measures is achieved for the criterion of statistical significance when a=0.01 and for the other two significance criteria when the parameters corresponding to the true density $\rho_{0}$, as expected. Comparing the three rankings for the best parameter choice of each criterion, it is observed that the statistical significance gives MCC values almost as high as the other two methods. This fact indicates the advantage of the statistical significance criterion, where the a priori knowledge of the true network density is not required. From this point on, all the presented results are for the criterion of statistical significance.

We here discuss the dependence of the measure accuracy on the coupling strength *C* and use as an example the system S1 with K=25 and n=2048. In Figure 5, the MCC for PMIME, RGPDC, RCGCI, and TE is given as a function of *C* for the three significance criteria.

The parameters in the criteria are α=0.05 for the statistical testing, the true density of the original network $\rho_{0}\;=\;48$, and the average magnitude threshold $th_{\rho_{0}}$ over all magnitude thresholds corresponding to the true density for the 10 realizations. For all significance criteria, the PMIME exhibits the best performance for large C>0.1 while TE has the highest MCC for small *C* at 0.05 and 0.1. Though S1 is a nonlinear system, the linear measures RCGCI and RGPDC (using the same dimension reduction step) are competitive and as good as or better than TE for large *C*. Thus, for this system and setting of *n*, *K* and large *C*, the rate of indirect (false) couplings found by the nonlinear bivariate measure TE is as large as or larger than the undetected nonlinear couplings from the linear measures RCGCI and RGPDC. This indicates that linear measures with dimension reduction may even perform better than nonlinear ones in settings of time series from nonlinear systems. It is noted that the coupling strength C=0.05 is very weak, and the dimension reduction methods PMIME, RCGCI, and RGPDC find no significant causal effects giving zero, which cannot change with any significance criterion. On the other hand, the small TE values for C=0.05 are still found significant at a good proportion giving rather large MCC at the level of 0.8.

### 3.3. Ranking of Causality Measures for Each Synthetic System

We derive summary results of all measures at each system over all coupling strengths *C* and for different number of variables *K* and time series lengths *n* where applicable. For this, we use the average score index $s_{i,j}$ for each measure *i* at each system *j* as defined in Section 2.4. In all results in this section, the statistical significance testing for α=0.05 has been used.

In Table 5, the average score $s_{i,j}$ for system S1 (coupled Hénon maps) is presented for the eight measures scoring highest at each scenario combining K=5 and K=25 with n=512 and n=2048.

In all scenarios of S1, the PMIME measure is found to have the best performance. In addition, the other measures of dimension reduction RCGCI and RGPDC reach highly ranked positions in all scenarios, especially in the case of large time series length. It is noted that these two measures are linear and they beat many other nonlinear measures showing the importance of proper dimension reduction. For small time series length (n=512), the information measures show better performance and it is again notable that the bivariate measures, such as TE, STE, and TERV, score higher than the corresponding multivariate measures, PTE, PSTE, and PTERV. Again, the explanation for this lies in the inability of the multivariate measures to deal with high dimensions if dimension reduction is not employed. Having even as low as three conditioning variables in the conditional mutual information used by these measures (in the case of K=25, the three more correlated variables in terms of MI to the driving variable are selected from the 23 remaining variables) does not provide as accurate estimates of the causal effects as the respective bivariate measures. These multivariate measures (along other multivariate measures of no dimension reduction) give non-existing causal effects even to the beginning and end of the ring, whereas the respective bivariate measures do not, and only estimate additionally indirect causal effects (results not shown here).

In Table 6, the average score for system S2 of coupled Mackey–Glass subsystems is presented for the eight measures scoring highest at each scenario combining K=5 and K=25 with Δ=100 and Δ=300, where Δ controls the complexity of each subsystem.

This system is comprised of highly complex systems with complexity increasing with Δ. For K=5 and regardless of Δ, the linear measures using dimension reduction RCGCI and RGPDC show the best performance, indicating again the importance of dimension reduction, here for oscillating complex systems. The PMIME scores slightly lower than these measures and, given that RGPDC scores equally high at different bands, together with RCGCI, they occupy the first eight places, so that the PMIME is not listed. For K=25, on the other hand, the PMIME scores much higher than the RCGCI and RGPDC measures and is in the first place for both Δ=100 and Δ=300. Apparently, the dimension reduction in the information measure of PMIME is more effective than in the VAR-based measure of RCGCI and RGPDC for larger *K*.

In Table 7, the average score for system S3 of the neural mass model is presented as for S2, but having as system parameter A=3.45,3.7, where the latter value indicates more clear oscillating behavior.

In all scenarios, GPDC shows the best performance. The RCGCI measure for A=3.45 reaches the next position, and RGPDC also reaches a high position on the ranking in the first three scenarios. The fact that GPDC scores higher than RGPDC also for K=25 indicates that, for this system and both *A*, the inclusion of all lagged terms in VAR of order p=5 or p=20 gives somehow better identification of the correct couplings after significance testing. This is so, due to the relative large length n=4096 of the time series that allows for the reliable estimation of the coefficients being as many as 20·25=500. For A=3.45, the PMIME does not score high as its sensitivity is comparatively small (fails to find significant proportion of true causal effects), whereas, for A=3.7, the PMIME is also among the first eight best measures. It is observed that, in all settings, the frequency measures, and particularly at low frequency bands, have the ability to identify the true causality interactions better than information and other measures. This is reasonable since this system is characterized by strongly harmonic oscillations.

For S4, no average results are shown as the system is run for only one scenario, and the ranking for this was shown in Table 4 and discussed earlier.

### 3.4. Overall Ranking of Causality Measures

For an overall evaluation of the causality measures, the average score $s_{i}$ over all systems and scenarios is computed for each measure *i*, as defined in Section 2.4. In Table 8, the ten measures with highest score $s_{i}$ are listed.

It is noted that, for each measure computed for varying parameters, such as the frequency bands for the frequency measures, only the one with the highest score is listed. The best performance is achieved by the three measures making use of dimension reduction, with the information measure PMIME scoring highest. It is noted that there is a jump in score from the third to the fourth place, showing the superiority of the measures of dimension reduction over the other measures. The remaining places in the list are dominated by the linear measures in the time and frequency domain. Comparing the frequency measures based all on the same VAR model, we note GPDC and even PDC scores higher than dDTF. As for the information measures, the bivariate measures TE and TERV score much higher than the multivariate respective measures (results not shown), indicating the inability of multivariate information measures to perform well unless an appropriate dimension reduction is applied.

## 4. Discussion

In this paper, a simulation study is performed for the estimation of causality networks from multivariate time series. For the network construction, Granger causality measures, simply termed here as causality measures, of different types were employed as information and model-based measures, measures based on phase, frequency measures, and measures making use of dimension reduction. These measures are applied to linear and nonlinear (chaotic), deterministic and stochastic, coupled simulated systems, to evaluate their ability to detect the existing coupled pairs of observed variables of these systems. We considered the nonlinear dynamical systems of coupled Hénon maps (S1), coupled Mackey–Glass subsystems (S2), the so-called neural mass model (S3), and a linear vector autoregressive process (VAR) of order 3 (S4). For systems S1, S2, and S3, we used K=5 and K=25 subsystems, whereas S4 was defined only on K=25 variables. For S2 and S3, we considered two regimes of different complexity for each system, controlled by a system parameter. For S1, S2, and S3, a range of coupling strengths *C* were designed covering the setting of none to weak and strong coupling. For S1, a small and a large time series length *n* were used. This design of the simulation aimed at testing the causality measures on different types of systems with respect to time (S1, S4 are discrete and S2, S3 continuous in time), low and high dimensional having K=5 and K=25, linear (S4) and nonlinear (S1, S2, S3), deterministic (S1, S2) and stochastic (S3, S4), and for a range of coupling strengths (S1, S2, S3). Thus, rather than concentrating on a particular system or type of systems, e.g., often met in EEG studies, we aimed at evaluating the performance of the causality measures on many different system settings. Measures that were best suited for strongly oscillating systems, such as the frequency measures, may not be appropriate for maps, and, on the other hand, information measures on ranks (such as TERV) that are more appropriate for discrete-time systems (maps) may not be appropriate for strongly oscillating signals. However, the evaluation showed that this was not the case, and, in the overall ranking, frequency measures dominated, but also TERV was included among the ten best.

The evaluation of the measures was based on the match of the causality network constructed from each measure to the original coupling network of the system generating the multivariate time series. For this, three significance criteria were used to transform the value of each causality measure, corresponding to a weighted network connection, to a binary value, corresponding to a binary connection. While the criteria of network density threshold and magnitude threshold are arbitrary and best results are only attained when the thresholds are given based on the knowledge of the original coupling network, the statistical testing, which does not require a priori information on the underlying system, was competitive and was further suggested as the criterion of choice to derive binary connections. Performance indices were computed checking the preservation of the original binary connections (true) in the causality binary network (estimated), and we used the Matthews correlation coefficient (MCC) to quantify best the matching performance of each causality measure as it weighs sensitivity and specificity.

We considered bivariate and multivariate causality measures, and a subset of three multivariate measures making use of dimension reduction. The first is the information measure of partial mutual information from mixed embedding (PMIME), which can be considered as a restriction of the partial transfer entropy (PTE) to the most relevant lagged variables. The other two measures are linear and they are both based on VAR model. The dimension reduction suggests fitting a sparse VAR rather than a full VAR. While the conditional Granger causality index (CGCI) is defined in the time domain on the full VAR, the restricted CGCI (RCGCI) is computed on the sparse VAR, and accordingly in the frequency domain the generalized partial directed coherence (GPDC) is modified using a sparse VAR to the restricted GPDC (RGPDC).

While linear models can be estimated sufficiently well in high-dimensional time series, provided the length of the time series is much larger than the number of the unknown model coefficients, the estimation of entropies, used in information measures, in high dimension is problematic, even when using the most appropriate estimate of nearest neighbors. To make a fair comparison between the multivariate information measure making use of dimension reduction (PMIME) to the other multivariate information measures in high-dimensional time series (here K=25), for the latter measures, we do not condition the causal relationship among the driving and response to all remaining variables (23 in our case), but rather select the three variables that are best correlated in terms of mutual information to the driving variable. In this way, we avoid to some degree the curse of dimensionality, but still the embedding is done separately for each of the five variables, i.e., the driving, the response and the other three variables, whereas, for the measures using dimension reduction, the embedding is built jointly for all variables selecting only the most appropriate lagged variables.

The evaluation of the causality measures showed differences in their performance in the different systems and their parameters (*n*, *K*, *C* and system parameter Δ for S2 and *A* for S3). For system S1, the measures making use of dimension reduction scored highest regardless of *n* and *K* with the PMIME attaining highest MCC for all but very small *C*. Bivariate information measures scored high here but only for small *n*. For system S2, the frequency measures were the most accurate at all frequency bands, especially for small *C* while for stronger couplings dimension reduction measures reached higher MCC. For high-dimensional time series (K=25), the PMIME again scored highest followed mainly by the linear dimension reduction measures for different parameter values. For system S3 characterized by strong oscillations, the frequency measures performed best occupying the highest ranks and only the PMIME entered the list of eight highest rankings for the system parameter A=3.7 for both *K*. For the linear VAR system S4, as expected, the linear measures with dimension reduction performed best and the respective linear measures of full dimension had also increased performance.

The conclusions of the simulation study on comparatively low-dimensional (K=5) and high-dimensional (K=25) time series from different systems are itemized as follows:The multivariate measures making use of dimension reduction (PMIME, RCGCI, RGPDC) outperform all other bivariate and multivariate measures.Among the dimension reduction measures, the information measure of PMIME is overall best but the overall score is slightly higher than that of the other two linear measures. Though the PMIME outperforms the other measures in the chaotic systems S1 and S2, for the strongly oscillating stochastic system S3 and the linear stochastic process S4, it scores lower than RCGCI and RGPDC.Linear measures, especially these applying dimension reduction, exhibited a competitive performance to other nonlinear measures also on nonlinear systems, such as S1, S2, and S3. This remark supports the use of linear measures (preferably with dimension reduction) to settings that may involve nonlinear relationships. Certainly, results still depend on the studied system.Though bivariate measures tend to identify causality relationships that are not direct, they do not fail in identifying non-existing causal relationships. The latter occurs when using multivariate measures without dimension reduction. Though this effect cannot be captured by standard performance indices used in this study, such as the sensitivity and specificity, it is a significant finding advocating against the use of multivariate measures, unless dimension reduction is applied.

Admittedly, the collection of causality measures is biased including all measures our team has developed. On the other hand, the collection is not comprehensive, leaving out nonlinear measures that are more difficult to implement and could not be found freely available when the study was initiated. It is noted that, initially, many connectivity measures that are not directional, especially these based on phases, were included, but they could not be fairly evaluated in the designed framework comparing the derived network to the original network of directed connections. The simulation study was conducted using four systems, leaving out other systems, such as the coupled Lorenz or coupled Rössler systems, as well as different coupling structures, such as the random (used here only in S4), small-world and scale-free, used by our team in other studies. Besides these shortcomings, we believe the current study can be useful for methodologists and practitioners to assess the strengths and weaknesses of the different causality measures and their applicability especially to high-dimensional time series.

## 5. Conclusions

A range of Granger causality measures were assessed comparing the estimated causality networks to the original networks of the systems generating the multivariate time series. In this simulation study, we used low and high dimensional coupled dynamical systems in discrete and continuous time, as well as deterministic and stochastic systems. The causality measures were bivariate and multivariate, in the time domain, such as model-based and information measures, the frequency domain, and the phase domain. In particular, linear model-based measures and information measures making use of dimension reduction were included. The evaluation of the matching of the causality networks estimated by the causality measures to that of the original system showed that the causality measures using dimension reduction preserve best the original coupling structure on both low-dimensional (5 variables) and high-dimensional (25 variables) multivariate time series. The study suggests the use of causality measures including dimension reduction in the causality and network analysis of time series from many variables or coupled subsystems, such as multi-channel electroencephalograms and financial markets.

## Figures and Tables

**Figure 1 entropy-21-01080-f001:**
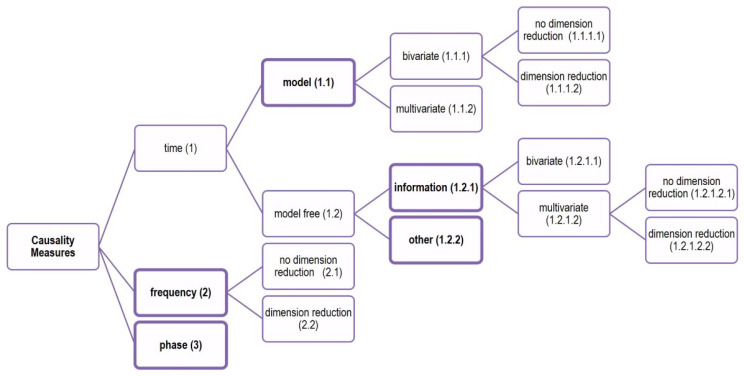
Tree structure for the types of causality measures. The five main classes are highlighted (frame box in bold). Each measure type in a box is given a code number, used as a reference in Table 1.

**Figure 2 entropy-21-01080-f002:**
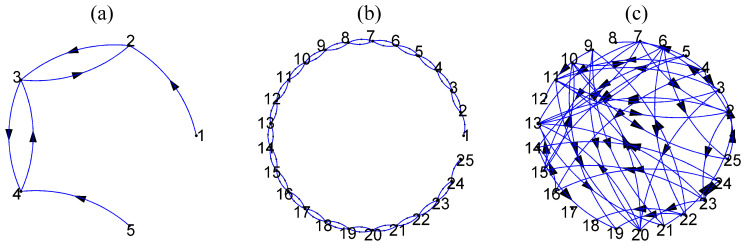
Coupling networks for S1, S2, S3 in (**a**) for K=5, in (**b**) for K=25, and in (**c**) for S4 and K=25.

**Figure 3 entropy-21-01080-f003:**
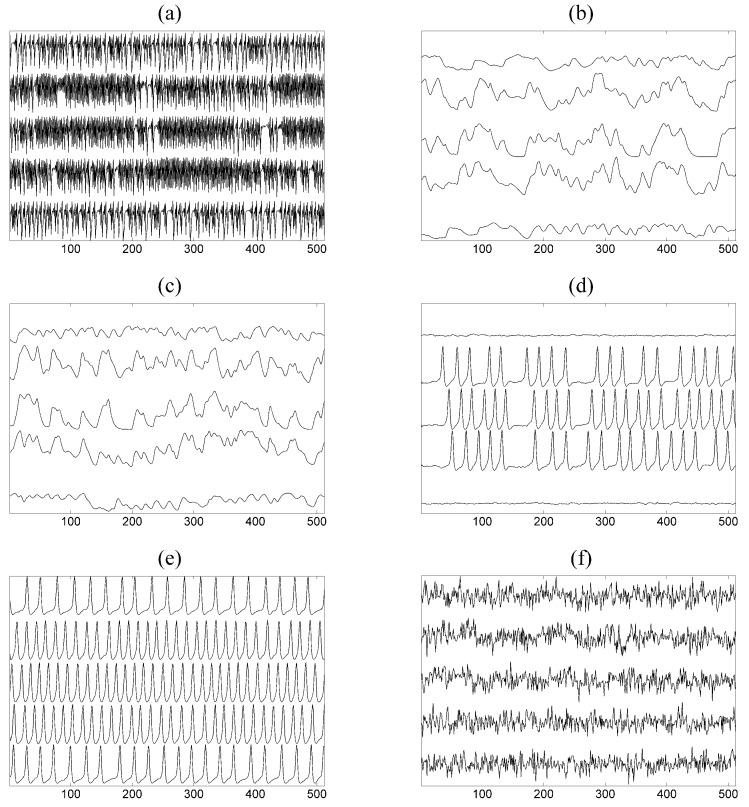
Time series for: (**a**) coupled Hénon maps for C=0.2; (**b**) coupled Mackey–Glass for Δ=100 and C=0.2; (**c**) coupled Mackey–Glass for Δ=300 and C=0.2; (**d**) neural mass for A=3.45 and C=80; (**e**) neural mass for A=3.7 and C=80; (**f**) VAR(3) model for C=0.23.

**Figure 4 entropy-21-01080-f004:**
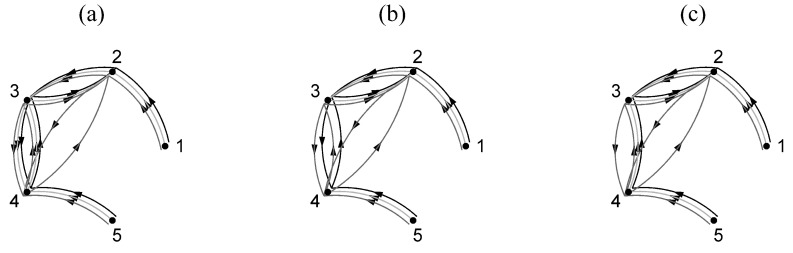
Binary networks of the measure of transfer entropy (TE) for S1 using (**a**) statistical significance test (light grey →α=0.01, black →α=0.05, grey →α=0.1), (**b**) density threshold (light grey →ρ=4, black →ρ=6, grey →ρ=8) and (**c**) magnitude threshold (light grey →$th_{4}$, black →$th_{6}$, grey → $th_{8}$)

**Figure 5 entropy-21-01080-f005:**
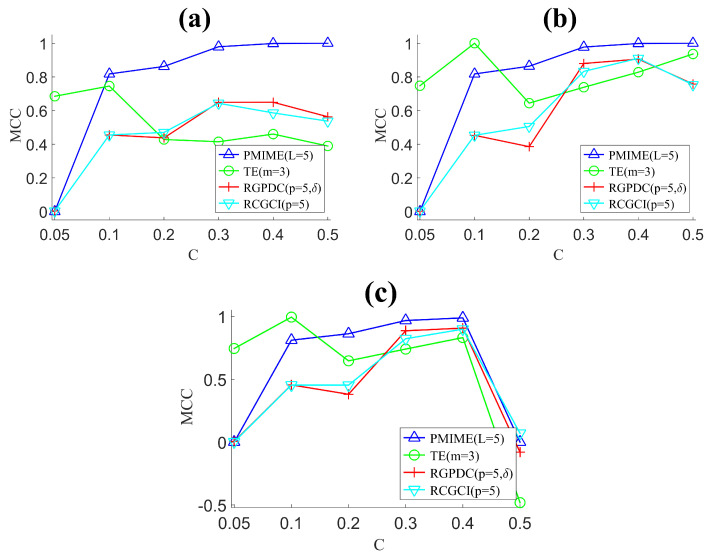
MCC of PMIME (L=5), TE (m=3), RGPDC (p=5,δ), RCGCI (p=5) as a function of the coupling strength *C* in system S1 of the coupled Hénon maps (K=25, n=2048) for the three significance criteria: (**a**) statistical testing at α=0.05, (**b**) true density threshold $\rho_{0}\;=\;48$ and (**c**) magnitude threshold $th_{\rho_{0}}$.

**Table 1 entropy-21-01080-t001:** List of causality measures organized in the five classes. The first column has the measure notation including the measure parameters, the second column has a brief description, the third column has the code of each measure denoting its type, and the fourth column has a related reference.

Symbol	Description	Type	Ref.
*Model-based*			
GCI(*p*)	Granger causality index, *p* is the VAR order, for Hénon maps p=2,5, for VAR process p=3,5, for the Mackey–Glass system and neural mass model p=5,10,20	1.1.1.1	[3]
CGCI(*p*)	Conditional Granger causality index	1.1.2	[22]
PGCI(*p*)	Partial Granger causality index	1.1.2	[72]
RCGCI(*p*)	Restricted Granger causality index	1.1.1.2	[43]
*Information*			
TE(m,τ)	Transfer entropy. Lag τ=1 for all systems, embedding dimension for Hénon maps m=2,3, for VAR process m=3,5, for Mackey–Glass system and neural mass model m=5,10,15	1.2.1.1	[27]
PTE(m,τ)	Partial transfer entropy	1.2.1.2.1	[73]
STE(m,τ)	Symbolic transfer entropy	1.2.1.1	[74]
PSTE(m,τ)	Partial symbolic transfer entropy	1.2.1.2.1	[75]
TERV(m,τ)	Transfer entropy on rank vectors	1.2.1.1	[76]
PTERV(m,τ)	Partial transfer entropy on rank vectors	1.2.1.2.1	[77]
PMIME(*L*)	Partial mutual information from mixed embedding, maximum lag for Hénon maps and VAR model $L_{max}\;=\;5$, for Mackey–Glass system and neural mass model $L_{max}\;=\;20$	1.2.1.2.2	[42]
*Frequency*			
PDC(p,i)	Partial directed coherence, *i* denotes the power band i=δ,θ,α,β,γ (relative proportion of the whole spectrum). For Hénon maps p=2,5, for the VAR process p=3,5, for the Mackey–Glass system and the neural mass model p=5,10,20	2.1	[24]
GPDC(p,i)	Generalized partial directed coherence	2.1	[78]
DTF(p,i)	Directed transfer function	2.1	[23]
dDTF(p,i)	Direct directed transfer function	2.1	[79]
GGC(p,i)	Geweke’s spectral Granger causality	1.1.1.1	[80]
RGPDC(p,i)	Restricted generalized partial directed coherence	2.2	[81]
*Phase*			
DPI	Phase directionality index	3	[30]
*Other*			
MCR(m,τ)	Mean conditional recurrence, *m* is the same as for the information measures	1.2.2	[26]
DED	Directed event delay	1.2.2	[82]

**Table 2 entropy-21-01080-t002:** Computation of the performance indices for the causality measure TE, where the binary causality network is derived using the statistical significance criterion for α=0.1.

	True Positive	True Negative
Positive Found	TP = 6	FP = 2
Negative Found	FN = 0	TN = 12
sens: 6/6=1		
spec: 12/14=0.86		
MCC: $\frac{6\cdot 12-2\cdot 0}{\sqrt{\left(6+2)\cdot (6+0)\cdot (12+2)\cdot (12+0\right)}}$=0.80		
FM: $\frac{2\cdot 6}{2\cdot 6+0+2}$=0.86		
HD: 2+0=2		

**Table 3 entropy-21-01080-t003:** The rankings of the eight best measures according to MCC for coupled Hénon maps (K=25, n=2048, C=0.2) are presented for the three binarization methods. The sensitivity (sens), specificity (spec), F measure (FM), and Hamming distance (HD) performance indices are also presented.

	Measure	sens	spec	MCC	FM	HD
statistical significance test (α=0.05)						
1	**PMIME**(L=5)	0.79	0.99	0.86	0.86	11
2	**RGPDC**(p=5,β)	0.86	0.85	0.49	0.49	84.5
3	**RGPDC**(p=5,α)	0.87	0.84	0.47	0.47	91.6
4	GPDC(p=5,β)	0.87	0.84	0.47	0.46	92.3
5	**RCGCI**(p=5)	0.92	0.81	0.47	0.45	106.1
6	**RGPDC**(p=5,γ)	0.86	0.83	0.46	0.46	95.8
7	GCI(p=5)	0.83	0.84	0.45	0.45	92.9
8	**RGPDC**(p=5,θ)	0.83	0.84	0.45	0.45	95.4
density threshold ($\rho_{0}\;=\;48$)						
1	**PMIME**(L=5)	0.79	0.99	0.86	0.86	10.9
2	TE(m=2)	0.68	0.97	0.66	0.68	28.8
3	TE(m=3)	0.67	0.97	0.64	0.67	30.2
4	PGCI(p=5)	0.6	0.96	0.56	0.6	36.8
5	GPDC(p=5,β)	0.59	0.96	0.56	0.59	37
6	GPDC(p=5,α)	0.57	0.96	0.54	0.57	38.8
7	**RGPDC**(p=5,β)	0.56	0.96	0.52	0.56	40
8	CGCI(p=5)	0.56	0.96	0.52	0.56	40
magnitude threshold ($th_{48}$)						
1	**PMIME**(L=5)	0.78	0.99	0.86	0.86	11
2	TE(m=2)	0.67	0.97	0.65	0.67	29.9
3	TE(m=3)	0.66	0.97	0.64	0.67	29.9
4	GPDC(p=5,β)	0.58	0.96	0.55	0.57	39.8
5	GPDC(p=5,α)	0.57	0.95	0.53	0.56	41.8
6	**RGPDC**(p=5,β)	0.54	0.96	0.51	0.55	40.4
7	PGCI(p=5)	0.50	0.96	0.51	0.52	40.6
8	**RGPDC**(p=5,α)	0.53	0.96	0.50	0.54	41.9

**Table 4 entropy-21-01080-t004:** The rankings of the eight best measures according to MCC for the system S4 of the VAR process (K=25, n=512, p=3) in conjunction with each significance criterion and its parameter. Three rankings are given in three blocks, one for each significance criterion and for three different choices of its parameter, where $\rho_{0}\;=\;48$ is the true density.

Statistical Significance Test				
	Measure	a=0.01	a=0.05	a=0.1
1	**RGPDC(p=3, α)**	0.944	0.868	0.861
2	**RGPDC(p=5, α)**	0.944	0.867	0.861
3	**RGPDC(p=3, θ)**	0.940	0.868	0.861
4	**RGPDC(p=5, θ)**	0.939	0.867	0.861
5	**RGPDC(p=3, δ)**	0.938	0.868	0.861
6	**RGPDC(p=5, δ)**	0.936	0.867	0.861
7	**RGPDC(p=5, β)**	0.933	0.867	0.861
8	**RCGCI(p=3)**	0.933	0.867	0.861
density threshold				
	Measure	0.6$\rho_{0}$	$\rho_{0}$	1.4$\rho_{0}$
1	**RCGCI(p=3)**	0.758	0.974	0.862
2	**RGPDC(p=5, α)**	0.758	0.972	0.862
3	**RCGCI(p=5)**	0.758	0.972	0.862
4	**RGPDC(p=3, α)**	0.758	0.972	0.862
5	**RGPDC(p=3, γ)**	0.755	0.972	0.862
6	**RGPDC(p=5, γ)**	0.755	0.972	0.862
7	**RGPDC(p=5, θ)**	0.785	0.972	0.862
8	**RGPDC(p=3, β)**	0.758	0.969	0.862
magnitude threshold				
	Measure	$th_{0.6\rho_{0}}$	$th_{\rho_{0}}$	$th_{1.4\rho_{0}}$
1	**RCGCI(p=3)**	0.751	0.979	0.868
2	**RGPDC(p=3, θ)**	0.753	0.976	0.868
3	**RGPDC(p=5, θ)**	0.753	0.975	0.869
4	**RGPDC(p=3, α)**	0.757	0.975	0.868
5	**RGPDC(p=3, δ)**	0.750	0.975	0.868
6	**RCGCI(p=5)**	0.752	0.974	0.868
7	**RGPDC(p=5, α)**	0.752	0.974	0.869
8	**RGPDC(p=5, δ)**	0.751	0.974	0.868

**Table 5 entropy-21-01080-t005:** The ranking of the eight best measures according to the score index for system S1 of coupled Hénon maps for all scenarios of number of variables *K* and time series length *n*.

K=5				K=25			
n=512		n=2048		n=512		n=2048	
Measure	Score	Measure	Score	Measure	Score	Measure	Score
**PMIME**(L=5)	0.87	**PMIME**(L=5)	0.84	**PMIME**(L=5)	0.92	**PMIME**(L=5)	0.91
TERV(m=3)	0.78	**RGPDC**(p=5,δ)	0.77	TE(m=2)	0.87	**RGPDC**(p=5,δ)	0.83
STE(m=3)	0.77	**RGPDC**(p=5,θ)	0.76	TE(m=3)	0.86	**RGPDC**(p=5,θ)	0.83
TE(m=3)	0.72	**RCGCI**(p=5)	0.76	**RGPDC**(p=5,β)	0.81	**RCGCI**(p=5)	0.82
**RCGCI**(p=5)	0.71	**RGPDC**(p=5,γ)	0.75	**RGPDC**(p=5,α)	0.81	**RGPDC**(p=5,β)	0.82
**RGPDC**(p=5,α)	0.71	**RGPDC**(p=5,β)	0.75	**RGPDC**(p=5,θ)	0.79	**RGPDC**(p=5,α)	0.81
CGCI(p=5)	0.70	PDC(p=5,α)	0.75	**RGPDC**(p=5,δ)	0.78	**RGPDC**(p=5,γ)	0.80
**RGPDC**(p=5,γ)	0.70	GPDC(p=5,α)	0.73	**RCGCI**(p=5)	0.77	CGCI(p=5)	0.76

**Table 6 entropy-21-01080-t006:** The ranking of the eight best measures according to the score index for system S2 of coupled Mackey–Glass subsystems for all scenarios of number of variables *K* and delay parameter Δ that controls the complexity of each subsystem.

K=5				K=25			
Δ=100		Δ=300		Δ=100		Δ=300	
Measure	Score	Measure	Score	Measure	Score	Measure	Score
**RGPDC**(p=20,α)	0.90	**RCGCI**(p=20)	0.88	**PMIME**(L=50)	1.00	**PMIME**(L=50)	0.95
**RGPDC**(p=20,δ)	0.88	**RGPDC**(p=20,γ)	0.86	PGCI(p=5)	0.87	**RCGCI**(p=5)	0.89
**RGPDC**(p=20,β)	0.88	**RGPDC**(p=20,δ)	0.86	**RGPDC**(p=20,γ)	0.85	**RGPDC**(p=5,δ)	0.88
**RGPDC**(p=20,γ)	0.88	**RGPDC**(p=20,α)	0.86	**RGPDC**(p=20,δ)	0.85	**RGPDC**(p=5,γ)	0.87
**RCGCI**(p=20)	0.86	**RGPDC**(p=20,θ)	0.85	**RGPDC**(p=20,β)	0.84	**RGPDC**(p=5,β)	0.86
**RGPDC**(p=20,θ)	0.86	**RGPDC**(p=20,β)	0.85	**RCGCI**(p=20)	0.84	**RGPDC**(p=5,α)	0.86
**RGPDC**(p=5,θ)	0.86	**RGPDC**(p=5,γ)	0.80	**RGPDC**(p=20,α)	0.83	**RGPDC**(p=5,θ)	0.85
**RGPDC**(p=5,δ)	0.84	**RCGCI**(p=5)	0.78	PGCI(p=20)	0.82	**RGPDC**(p=20,α)	0.81

**Table 7 entropy-21-01080-t007:** The ranking of the eight best measures according to the score index for system S3 of the neural mass model for all scenarios of number of variables *K* and oscillation controlling parameter *A*.

K=5				K=25			
A=3.45		A=3.7		A=3.45		A=3.7	
Measure	Score	Measure	Score	Measure	Score	Measure	Score
**RCGCI**(p=20)	0.88	GPDC(p=20,θ)	0.94	GPDC(p=5,θ)	0.83	GPDC(p=20,θ)	0.92
GPDC(p=20,θ)	0.88	PDC(p=20,θ)	0.88	GPDC(p=20,θ)	0.82	CGCI(p=20)	0.90
**RGPDC**(p=20,α)	0.88	**PMIME(L=20)**	0.80	**RCGCI**(p=20)	0.81	dDTF(p=20,δ)	0.89
**RGPDC**(p=20,γ)	0.87	**RGPDC**(p=5,α)	0.78	**RGPDC**(p=20,δ)	0.79	GPDC(p=5,α)	0.87
PGCI(p=20)	0.87	**RCGCI**(p=5)	0.78	**RGPDC**(p=20,β)	0.79	**PMIME(L=20)**	0.87
CGCI(p=20)	0.87	**RGPDC**(p=5,α)	0.77	**RGPDC**(p=20,θ)	0.79	PDC(p=5,α)	0.82
**RGPDC**(p=20,θ)	0.86	**RGPDC**(p=20,θ)	0.77	CGCI(p=20)	0.78	PDC(p=20,θ)	0.80
**RGPDC**(p=20,δ)	0.85	**RGPDC**(p=20,γ)	0.77	PDC(p=20,θ)	0.78	GPDC(p=20,α)	0.80

**Table 8 entropy-21-01080-t008:** Average score index over all systems and scenarios.

Measure	Score
**PMIME**	0.80
**RGPDC**	0.79
**RCGCI**	0.78
GPDC	0.63
CGCI	0.61
PGCI	0.61
PDC	0.56
TE	0.51
dDTF	0.50
TERV	0.46

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
