# Peer review of "Evaluation of Granger Causality Measures for Constructing Networks from Multivariate Time Series"

_entropy, 2019, doi:10.3390/e21111080_

Round 1

Reviewer 1 Report

The paper is interesting and well written. I have the following comments:

-The definition of causality that is referred to as “Granger Causality” in literature was in fact first defined by Wiener (1956) and then it was promoted and developed by Granger (1969) . Thus, perhaps instead of calling it “Granger Causality” a better name would be “Wiener -Granger Causality”.

-The paper is missing other contributions within this context among others by Sims (1972).

-It is widely agreed in the literature that allowing for asymmetric causal impacts accords well with reality. There are asymmetric causality tests suggested by Hatemi-J (2012) that might be useful for this purpose.

-Testing for causality is based on a dynamic model in which the lag order needs to be determined appropriately since the entire inference is based on the selected lag order. More details about how the lag order should be chosen within this context for practitioners might be useful.

-Financial data is usually not normally distributed and the volatility is time varying. How do the authors tackle this issue?

-Is the issue of non-stationarity (i.e. integrated variables) not relevant in this case?

-Is there any consumer friendly software component that can be used by the practitioners in order to implement the suggested method? If yes, mentioning it in the paper would be useful.

References

Wiener, N. (1956) The theory of prediction. In: Beckenbach, E. (Ed.), Modern Mathematics for Engineers. McGraw-Hill, New York.

Sims, C.A., (1972) Money, Income and Causality, American Economic Review, 62, 540-552.

Hatemi-J, A. (2012) Asymmetric Causality Tests with an Application, Empirical Economics, 43(1), 447-456.

Reviewer 2 Report

This paper focuses on the estimation of causality networks from multivariate time series. Different types of causality measures are used in this study. The paper presents enough results and the presentation is pretty clear.

I suggest minor revision before considering for publication. My only main concern is the following. Since the paper includes quite a few measures. I would suggest adding more discussions for clarifying the underlying reasons of the discrepancy in the results based on different measures. The authors have presented some reasons but going into more depth would be helpful.

Reviewer 3 Report

The work is very interesting and there are many applications. The simulations and results are comprehensive to understanding.

My main concerns are:

in the introductions section, it is not clear the paper's contributions. Please  clearly point it and matches this in conclusion section; the research area of causality networks has many more aspects than the measure of individual causal multivariate signals.

Many aspects of the causal network can be found in the recent work of J. Pearl. Please check their references for improving your work. One of the main question about inferring the causal network in the ability to make questions around the model, or, counterfactual reasoning. 

The discussion in pg 2. 2nd paragraph must include other topology like (z-> x and z-> y) and (x->z and y->z). Please improve it. 
